# Local Therapies and Modulation of Tumor Surrounding Stroma in Malignant Pleural Mesothelioma: A Translational Approach

**DOI:** 10.3390/ijms22169014

**Published:** 2021-08-20

**Authors:** Daniela Lisini, Sara Lettieri, Sara Nava, Giulia Accordino, Simona Frigerio, Chandra Bortolotto, Andrea Lancia, Andrea Riccardo Filippi, Francesco Agustoni, Laura Pandolfi, Davide Piloni, Patrizia Comoli, Angelo Guido Corsico, Giulia Maria Stella

**Affiliations:** 1Cell Therapy Production Unit-UPTC and Cerebrovascular Diseases Unit, Fondazione IRCCS Istituto Neurologico Carlo Besta, 20133 Milan, Italy; daniela.lisini@istituto-besta.it (D.L.); sara.nava@istituto-besta.it (S.N.); simona.frigerio@istituto-besta.it (S.F.); 2Unit of Respiratory Diseases, Department of Medical Sciences and Infective Diseases, IRCCS Policlinico San Matteo Foundation and University of Pavia Medical School, 27100 Pavia, Italy; sara.lettieri01@universitadipavia.it (S.L.); accordino.giu@gmail.com (G.A.); l.pandolfi@smatteo.pv.it (L.P.); d.piloni@smatteo.pv.it (D.P.); a.corsico@smatteo.pv.it (A.G.C.); 3Unit of Radiology, Department of Intensive Medicine, IRCCS Policlinico San Matteo Foundation and University of Pavia Medical School, 27100 Pavia, Italy; c.bortolotto@smatteo.pv.it; 4Unit of Radiation Therapy, Department of Medical Sciences and Infective Diseases, IRCCS Policlinico San Matteo Foundation and University of Pavia Medical School, 27100 Pavia, Italy; a.lancia@smatteo.pv.it (A.L.); a.filippi@smatteo.pv.it (A.R.F.); 5Unit of Oncology, Department of Medical Sciences and Infective Diseases, IRCCS Policlinico San Matteo Foundation and University of Pavia Medical School, 27100 Pavia, Italy; f.agustoni@smatteo.pv.it; 6Cell Factory and Pediatric Hematology-Oncology Unit, IRCCS Fondazione Policlinico San Matteo, 27100 Pavia, Italy; p.comoli@smatteo.pv.it

**Keywords:** mesothelioma, micro-environment, local therapy, advanced cell therapies

## Abstract

Malignant Pleural Mesothelioma (MPM) is a rare and aggressive neoplasm of the pleural mesothelium, mainly associated with asbestos exposure and still lacking effective therapies. Modern targeted biological strategies that have revolutionized the therapy of other solid tumors have not had success so far in the MPM. Combination immunotherapy might achieve better results over chemotherapy alone, but there is still a need for more effective therapeutic approaches. Based on the peculiar disease features of MPM, several strategies for local therapeutic delivery have been developed over the past years. The common rationale of these approaches is: (i) to reduce the risk of drug inactivation before reaching the target tumor cells; (ii) to increase the concentration of active drugs in the tumor micro-environment and their bioavailability; (iii) to reduce toxic effects on normal, non-transformed cells, because of much lower drug doses than those used for systemic chemotherapy. The complex interactions between drugs and the local immune-inflammatory micro-environment modulate the subsequent clinical response. In this perspective, the main interest is currently addressed to the development of local drug delivery platforms, both cell therapy and engineered nanotools. We here propose a review aimed at deep investigation of the biologic effects of the current local therapies for MPM, including cell therapies, and the mechanisms of interaction with the tumor micro-environment.

## 1. Introduction

Malignant pleural mesothelioma (MPM) is a fatal asbestos-related malignancy which originates from the mesothelial layer which coat the pleural space. Despite asbestos bans, MPM incidence in Europe and Japan is still rising and it is almost reaching its peak, predicted in 2020 and 2025, respectively. In early disease stages, multimodal therapeutic approaches, encompassing surgery, radiation therapy and chemotherapy can be available at least in patients featuring good performance status. In advanced MPM cases the only option is palliative treatment. Several chemotherapeutics have already been evaluated, in absence of clear clinical efficiency [1]. Modern targeted therapies that have shown benefits in other human tumors have so far failed in MPM [2]. Thus, MPM has been listed an orphan disease by the European Union (EU). The latter assures that the disease is enclosed into the European Reference Network (ERN) for the lung that is dedicated to rare respiratory diseases, which has the main aims to: facilitate improvements in access to diagnosis and delivery of high quality, accessible and cost-effective healthcare for people living with rare diseases, and act as focal points for medical training and research, information dissemination and evaluation, especially for rare diseases (website at ERN-LUNG | Rare Respiratory Diseases). In addition to being a rare disease, MPM can also be classified as an orphan disease, poorly interesting for the pharmaceutical industry that does not invest sufficiently in this pathology. For patients that do not respond to the first-line chemotherapy, a second-line chemotherapy approach can be considered. However, there are no approved agents. Thus, MPM remains a disease setting to test new agents. Consequently, patients should be encouraged to participate in clinical trials [3]. When a clinical trial is not available, single-agent chemotherapy could be considered for fit patients, although the panel was not unanimous about this issue. Best supportive care remains a valid option. Among biologic agents that have shown unsuccessful results against MPM there are tyrosine kinase inhibitors and the PIK3CA-mTOR inhibitor, everolimus [4,5,6]. Furthermore, the first data from immunotherapy in MPM have been disappointing [7], with the recent exception that anti-PD-1 nivolumab alone in combination with the anti-CTLA-4 ipilimumab has shown good activity in absence of relevant toxicities, in 125 MPM patients featuring progressive disease after conventional chemotherapy [8,9]. In 2017, a phase 2b, multicenter, randomized, double-blind, controlled trial investigated tremelimumab, a CTLA-4 inhibitor, as a second/third line approach after disease relapse in a cohort of 569 patients, but no significant impact on overall survival (OS) was reported [10]. Subsequent translational studies confirmed that MPM is characterized by an immunosuppression-oriented stromal context which is coherent to relatively low response rates to checkpoint inhibitors. Thus, novel and targeted immunotherapeutic approaches are now under investigation and development.

The poor therapeutic results that impact MPM patient outcomes could be reasonably ascribed as both due to drug inability to reach the site of disease and to the difficulty in reaching adequate intracellular drug concentrations, due to excessive systemic side effects. MPM is characterized by rapid and diffuse local growth, whereas distant metastases rarely appear and arise in advanced phases of disease development. Thus, the pleural space, despite the physiological draining capacity of the membrane that covers it, represents an ideal space for the local chemotherapy treatment in case of neoplastic transformation. Moreover, growing evidence suggests that asbestos-induced inflammation is at the basis of the neoplastic transformation of mesothelial cells and that this unique tumor micro-environment (TME) is involved in the induction of resistance to the applied therapies [2,11]. Among the most promising therapeutic strategies that are under investigation, there is the local delivery of biologic and chemical agents. These approaches exploit the typical progression pattern of MPM which is most often localized, whereas distant spreading rarely occurs. There is a strong rationale for development of local delivery therapeutic platforms, mainly in case of a rare disease such as MPM. The first is that local delivery of an antineoplastic drug allows an increase in drug concentration near the tumor mass and reduces the risk of drug inactivation before reaching it. Secondly, the drug bioavailability is increased, and its higher activation also assures a deeper and more profitable interaction with the specific tumor surrounding stroma. Furthermore, local treatments reduce toxic effects on normal non-transformed cells and therefore of systemic chemo-toxicity because of much lower drug doses than those used for systemic chemotherapy. Several strategies are under development encompassing surgery, ionizing radiation, biologic agents and cells and nanoparticles. This present work aims at summarizing the novel approaches for local therapy in MPM, with a primary focus on the rationale of this approach, which assures a closed contact with the immune/inflammatory and vascular processes which specifically characterize the MPM stroma. From this perspective, advanced cell therapies, which ultimately aim at increasing the number of effector cells against the tumor, might play a substantial role which is favored by their dynamic interaction with tumor stroma.

## 2. Targeting MPM Stroma by Local Approach

### 2.1. Biologic Frame and Rationale

Growing evidence demonstrates that MPM is characterized by specific features of the micro-environment that hamper its pharmacological targeting. Indeed, MPM is not made merely of transformed cells, but a dynamic cross-talk with surrounding stroma is responsible for tumor onset and progression. Every tumor (and, thus, MPM) is made with malignant cells plus stroma: the main and specific issue is related to the role played by asbestos (rather than activation of specific oncogenes) in both driving malignant transformation and modulating tumors surrounding stroma [2,12]. Although a deep description of MPM surrounding stroma goes beyond the scope of this review and can be available in detail in recently published literature, e.g., [2,12,13,14,15], it is relevant to underline that the chronic inflammatory response to asbestos leads to the generation of a specific and heterogeneous stroma which sustains malignant transformation [16]. In detail, it determines an increased production of reactive oxygen species (ROS) and free radicals which recall inflammatory cells [17,18]. The inflammatory process is exacerbated by the activation of macrophages that are stimulated by the release of high-mobility group box 1 protein (HMGB1) and by secretion of TNF-alpha and other inflammatory cytokines in the intercellular spaces [19]. Overall, these processes reduce the peritumoral immunity milieu [20,21]. It is constituted by different cell types as immunosuppressive cells, such as type 2 tumor-associated macrophages and T regulatory lymphocytes, and several immunosuppressive factors, among which is tumor-associated PD-L1 [22]. The genetic asset of the disease also modulates peritumor micro-environment. For instance, it has been recently reported that p14/ARF-negative tumors display an immune micro-environment which is less sensitive to immune checkpoint inhibition, being associated with low PD-L1 and CD4 expression, and high CD163 percentage [23]. Frequent somatic alterations involve the *BAP1*, *NF2* and *CDKN2A* genes and leads to a significant enrichment of tumor suppressor genes. The MPM clonal evolution shapes the MPM micro-environment and modulates immunosurveillance, since neoantigens derived during clonal evolutionary trajectory can modulate the composition of the micro-environment by regulating the infiltration levels of Tregs, CD8+, the HLA system [24]. Notably, the characterization of the immune micro-environment has been significantly associated with patient prognosis [25], being those infiltrates featuring by low CD4^POS^ lymphocytes and high CD8^POS^ and high PD-L1 expression associated with poor patient survival [26,27,28]. Among the factors which regulate T cell activity, hypoxia is a main driver and promotes tumor cell growth and aggressiveness through the increased expression of several molecules such as hypoxia-inducible factor 1 (HIF1α/2α), CD44 and Oct4, Bcl2, E-cadherin, vimentin and key nutrients such as glucose transporter 1 (Glut1) [29]. Moreover, hypoxia, by enhancing HIF1α-expression, increases PD-L1 expression in animal models [30]. HIF1α expression is also associated with suppression of T-cell proliferation in mice [31]. In conclusion, the fibroinflammatory stroma typical of MPM, beyond impeded drug penetration in the tumor mass, can contribute to chemoresistance by stimulating cancer cells growth, invasion and angiogenesis, and inducing an immunosuppressive phenotype [32]. This observation sustains a strong rationale according to which the micro-environment is becoming an appealing actionable target. In this perspective, mesothelioma has two potential advantages. First, relatively specific markers have been identified. Second, MPM may provide an opportunity to use local therapy by intrapleural or intra-tumoral injection.

The specific pattern of growth of MPM could become an Achilles heel. An overview on the strategies that have been developed and/or are under investigation for local therapy in MPM is summarized in Figure 1. Many cellular and intracellular pathways have been exploited for therapeutic purposes in cancer and in MPM as well. In brief, cell surface targets include receptor tyrosine kinases involved in cell proliferation, invasion and angiogenesis, such as VEGFR, whose activation is blocked by the antibody Bevacizumab or by various tyrosine kinases inhibitors such as nintedanib, semaxinib, cedinarib, sorafenib and sunitinib, and immune checkpoint inhibitors such as PD-1/PDL-1 or CTLA-4 axis [33,34,35]. PD-1 is expressed on the surface of T-lymphocytes, B lymphocytes and NK cells. PD-1 ligand (PDL-1) is expressed both in hematopoietic and non-hematopoietic cells, including mesothelioma cells. PD-1/PDL-1 signaling plays a crucial role in tumorigenesis, inducing resistance against T cell mediated killing and promoting cancer immune escape [36,37]. PD-1 inhibitors such as pembrolizumab and nivolumab, and PDL-1 inhibitors such as durvalumab and atezolizumab, restore antitumor activity of T cells within tumor micro-environments. Cytotoxic T-Lymphocyte Antigen 4 (CTLA-4) is expressed in T cells and acts by blocking their activity through the interaction with CD80 or CD86 expressed on antigen presenting cells and cancer cells. The blockade of CTLA-4 by monoclonal antibodies such as ipilimumab and tramelimumab promotes T cell killing activity against cancer cells. A combination of PD-1/PDL-1 and CTLA-4 blockade shows a synergistic effect [38,39,40]. Intracellular pathways include apoptotic regulators such as PI3K/AKT/mTOR pathway, the target of mTOR inhibitor Everolimus, and epigenetic modifiers of histones such as HDAC inhibitor Belinostat [41]. Novel therapeutic strategies have been developing. Pegylated adenosine deaminase reduces extracellular adenosine concentrations, which promotes cancer growth within micro-environments, and induces apoptosis of MPM cells [41]. Mesenchymal stromal cells (MSCs), after in vitro expansion and manipulation to make them able to deliver antineoplastic agents, can exert anticancer activity through their secretoma, the release of a great number of soluble factors and their differentiation potentialities [42]. Adoptive cell therapy includes vaccination with dendritic cells (DCs) loaded with Tumor Associated Antigens (TAAs), which prime antigen-specific cytotoxic T cells and activate natural killer cells; engineer T cells to express chimeric antigen receptors (CARs) against targets such as mesothelin, FAP, MET and pan-ErbB; oncolytic viral immunotherapy, which uses nonpathogenic oncolytic viruses to infect and lyse cancer cells and to stimulate a robust immune anti-tumor response [2,43].

Essentially the tools in clinical use to locally vehicle anticancer drugs are cellular-derived or biologically compatible synthetic compounds such as polymer drug conjugates, nanoparticle and liposomal systems, and transdermal drug delivery patches. These novel technologies have made possible the development and use of new anticancer agents as well as a more a safer and more efficient profile of standard chemotherapy. Furthermore, local administration of anticancer therapies combing immune, chemo, radiotherapy and small agents can assure synergistic antiproliferative effects. In this perspective, the heterogeneity of MPM surrounding stroma could be exploited to obtain synergistic effects with conventional treatments.

It should be also noted that a tumor micro-environment is also modulated by physical forces. In a context of local drug delivery, they could play a crucial role and should be kept under consideration. They control trans-vascular and interstitial drug transport as well as intercellular cross-talk by modulating the interaction between invading cancer cells and their 3D micro-environments. Interestingly—with respect to MPM—it has been reported that Caveolin 1 (CAV1) acts as a multifunctional scaffolding protein which is involved in cancer growth and progression, modulating tissue responses through architectural regulation of the micro-environment. Caveolae and their components act as modulators of biomechanics and ECM–cell interactions [44,45].

### 2.2. Imaging as a Tool to Assess Tumor Micro-Environment

Images are more than pictures: they are data [46]. A decade ago this new approach to images created a completely new field of research called radiomics. Radiomics extracts large amounts of features from biomedical images using data-characterization algorithms [47]. These features, named radiomic features, can be used in treatment selection or as outcome prediction biomarkers.

Overall, the complex interaction between drugs and the local immune-inflammatory micro-environment which modulates the subsequent clinical response had already been explored with radiomics in several different scenarios, in particular for immune (PD-1, PD-L1) markers [48]. This approach to assess tumor-infiltrating CD8 cells and responses to anti-PD-1 or anti-PD-L1 immunotherapy has not been yet applied to MPM. Nonetheless, there have been several successful reports of the application of radiomics to different aspects of MPM management (e.g., plaque characterization) [49]. Local therapies and modulation of tumors surrounding stroma will most likely be one of the next research fields for radiomics in MPM. Although chest magnetic resonance (MR) is not routinely used to evaluate MPM, this imaging has great potentiality in evaluating local therapy responses due to its functional insights (Figure 2). Diffusion-weighted imaging (DWI) allows patient-tailored care in MPM: it has already been used to discriminate between long- and short-term overall survivors [50]. Several new tools have been developed in the last year for a multi-parametric evaluation of tumor lesions (e.g., Dynamic Contrast-Enhanced MR Imaging, Chemical Exchange Saturation Transfer Imaging) [51]. IVIM-based perfusion MRI, which does not require contrast agents, is gaining momentum, especially for oncologic applications [52,53]. Intravoxel Incoherent Motion (IVIM) refers to translational movements which, within a given voxel and during the measurement time, present a distribution of speeds in orientation and/or amplitude. IVIM can evaluate micro-vessel perfusion in vivo using quantitative parameters obtained from the multi-b-value DWI of a double-exponential decay model. So far, no attempt to evaluate a possible correlation between neo-angiogenesis (VEGF-VEGFR) markers and IVIM parameters for MPM has been reported.

## 3. How to Local Target MPM: Where We Are Going

### 3.1. Nanoparticles (NPs) as Novel Promise for MPM

The application of nanotechnology in medicine opens incredible opportunities in cancer treatment. Given the chemical and physical properties of NPs, they allow: (1) to target specific cell population exploiting both passive targeting by enhanced permeability and retention effects, and active targeting by modifying the surface of nanoparticles with moieties specific only for cancer cells; (2) to increase drugs concentration in tumor bulk reducing side effects; (3) to allow new route of administration for those drugs already used in clinics, increasing their bioavailability. Usually, in cancer treatment the preferentially route of NPs administration is the intravenous injection. However, regarding MPM or other lung diseases, the best approach to administer therapies is the local treatment through aerosol or pleural injection.

The most used type of NPs for MPM treatment are liposomes. This is because these kinds of nanovehicles are composed of a mixture of lipids and/or cholesterol, giving the property to be highly biocompatible. Moreover, liposomes give the opportunity to deliver both hydrophobic and hydrophilic compounds, but also genetic material. In MPM treatment, liposomes are exploited to deliver chemotherapeutics, such as doxorubicin [54,55], or pemetrexed [56,57], with the main goal to increase the concentration of drugs in the peritoneal cavity, to avoid fast degradation of drug molecules, and to have the possibility to prolong the release of the drug. For example, Ando et al. demonstrated that an injection into the pleural cavity of pemetrexed loaded into liposomes had more suppressive effect on tumor growth in an orthotopic mesothelioma mouse model, compared to free pemetrexed. Other important studies using liposomes concerns their biodistribution after pleural injection [58,59]. Marazioti et al. conducted an animal study by injecting in a pleural cavity different types of liposomes labeled with DiR, comparing their accumulation in normal mice to MPM-bearing mice. Authors demonstrated that several parameters affect liposomes retention, such as size and coating surface with polymers, e.g., (PEG), showing that small liposomes or PEGylated ones had higher retention. Interestingly, the pattern of liposome clearance from the pleural cavity was the same comparing normal mice to MPM bearing ones [59].

Not only liposomes but also other types of nanoparticles are being studying for MPM treatment, such as gold nanoparticles [60] or pH-responsive polymeric (expansile) nanoparticles [61,62]. Additionally, for these kinds of nanoparticles, the potentiality of using nanoparticles was demonstrated. Very interesting are the results obtained by Schulz et al. that associated chirurgical resection with treatment with paclitaxel-loaded expansile nanoparticles. Comparing treatments with paclitaxel alone or loaded into nanovehicles, the authors demonstrated that only nano-formulated paclitaxel is able to significantly extend animal survival compared to those animals treated only with surgical resection or with the addition of free paclitaxel. This gain of efficacy was due to the property of nanoparticles to prolong the release of drugs in the intraperitoneal cavity. This point is crucial in a type of cancer such as MPM, where after surgical resection it can remain microscopic in residuals, leading to a disease recurrence.

As mentioned above, the use of NPs allows the targeting of specific cell populations with specific moieties. MPM researchers have exerted the high expression by MPM cells of CD146 [60] and CD44 [63], two membrane proteins significantly relevant for the diagnosis and prognosis of MPM. Both papers show the high ability of targeted-NPs to be internalized specifically by MPM cells by in vitro experiments; Sakurai et al. suggested that targeted NPs tended to accumulate inside tumor bulk more than non-targeted NPs after pleural injection by in vivo experiments [59].

In conclusion, despite the relative novelty of this field, nano-based approaches could be a very promising option for MPM patients, as adjuvant therapy after surgical resection to avoid the relapse of cancer. However, it is very important to study the fate of NPs after pleural injections for every type of material or modification done to NPs, since Marziaroti et al. demonstrated that changing the formulation of liposomes or size could also change the biodistribution and clearance of liposomes [59].

### 3.2. Advanced Cell Therapy

#### 3.2.1. Adoptive Cell Therapy for MPM

Adoptive transfer of ex vivo cultured and expanded Tumor Infiltrating Lymphocytes (TILs) isolated from a resected tumor specimen has been used from 1988 for the treatment of different tumor types. The treatment with TILs demonstrated a significant disease regression only in melanomas [64], even if the median duration of response was only four months due to immune tolerance and tumor escape. Today, TILs represent an experimental treatment not used in routine clinical practice. TILs have not been successfully used against other cancers; moreover, TILs are limited by small numbers of invasive lymphocytes and lack of significant innate anti-tumor immunity enhancement [65]. On the contrary, Dendritic Cells (DCs) are widely used as adoptive cell therapy in different cancers, such as glioblastoma, melanoma, ovarian cancer and also MPM [66,67,68]. Vaccination with DCs is used to initiate an anti-tumor immune response. Immature DC are normally present in the tissue micro-environment and become activated when they meet foreign pathogens. This activation follows stimulation by exogenous signals via pattern recognition receptors such as toll-like receptors (TLR), and stimulates the DC migration to the draining lymph node and the presentation of the processed epitopes to T cells. During the T cell activation, DC secrete different cytokines and stimulate the immune responses toward TH1 and TH2. Due to their characteristics, DC have been used as vaccine platforms to induce anti-tumor immune responses via cytotoxic T lymphocytes [69]. To evade the TME immunosuppressive activity, DCs can be activated and loaded with selected Tumor Associated Antigens (TAAs) or whole tumor lysate in vitro. Three generations define the evolution of the DCs-therapy. In the first one, monocytes isolated from peripheral blood were cultured with GM-CSF and interleukin (IL) 4. This allows the differentiation of DCs to immature monocyte-derived DCs. The latter were loaded with TAAs or tumor lysate and re-injected without any further stimulation into the patient. Second-generation DCs-therapy provided additional stimulation of moDCs in vitro by adding a maturation/activation cocktail encompassing cytokines and immune stimulants, among which were poly I:C, TLR ligands and prostaglandin E2. Next generation DCs-therapy is based on the use of naturally occurring DCs (nDCs), which are directly purified from peripheral blood. Then, they are loaded with TAAs or tumor lysate during an in vitro step. Once activated, they are used. Overall, this approach allows better culture performance and lower manufacturing costs [70,71,72].

In MPM, initial approaches used autologous tumor lysate loaded DCs and have shown unexpected and persistent clinical responses with significantly increased OS (with a maximum of 66 months) after the treatment. Although these are relevant data, this approach is impaired by two important disadvantages: it is time consuming and may not assure the achievement of quality standards for DCs therapy. Allogenic tumor lysates may be of help in bypassing this obstacle, as recently showed in a Phase I clinical trial MesoCancerVa (NCT02395679). In this trial, no dose-limiting toxicities were registered, and radiographic responses were observed. The median progression free survival (PFS) was 8.8 months and the median OS was not reached at a median follow-up of 22.8 months. In a follow up analysis of the peripheral blood T cell receptor β (TCRβ) chain repertoire of nine MPM patients before and five weeks after the start of DCs-based immunotherapy, it was found that clinical responses to DCs-mediated immunotherapy was related to both the pre-existing TCRβ repertoire of total CD3 + T cells and to therapy-induced changes, mainly expanding PD1+CD8+-T cell clones. Thus, the TCRβ profiling could potentially allow to identify and select those subsets of MPM patients that could really benefit from DCs-based immunotherapy. These promising results will be further assessed by the Phase II/III DENIM trial (NCT03610360) which aims to recruit *n* = 230 patients to examine the OS in patients treated with DCs loaded with this allogeneic tumor cell lysate, as a maintenance treatment after chemotherapy [73].

Another adoptive cell therapy approach in MPM treatment involves the use of T cells genetically engineered to express a chimeric antigen receptor (CAR) able to recognize a cell-surface antigen and kill cancer cells (CAR-T) [74].

The potential to use CAR-T therapy in MPM has been extensively evaluated, and pre-clinical models using various targets, among which are mesothelin (MSLN), Fibroblast Activation Protein (FAP), Met Proto-oncogene (cMET), pan-ErbB and others, have been tested. The biggest potential limitation of the use of CAR-T therapies in solid tumors could be T-cell exhaustion. However, recent data have pointed out that exposure to checkpoint inhibitors may improve the potency of CAR-T cell therapies, and other treatments such as co-stimulation induction and cytokine-based approaches may also have a role [75,76]. Nevertheless, the successful results obtained by CAR T-cell therapy in onco-hematology cannot be easily translated in the context of solid cancers, as in MPM, since some issues are not fully addressed and might impair CAR-T cell function, namely tumor antigen heterogeneity, the immunosuppressive tumor surrounding stroma, and the inhibition of immune cell trafficking.

#### 3.2.2. Drug Loading and Drug Delivery by Mesenchymal Stromal Cells (MSCs) and Their Extracellular Vesicles

Another potential therapeutic strategy is the use of Mesenchymal Stromal Cells (MSCs). MSCs are characterized by their ability to self-renew and differentiate into tissue-specific specialized cells. According to the International Society for Cellular Therapies (ISCT), minimal criteria to define multipotent MSCs are as follows: (1) plastic adhesion capacity in standard culture conditions, (2) surface expression of CD73, CD90, CD105, CD166, CD44 and CD29 markers and absence of CD14, CD34, CD45 and CD31 (3) differentiation capacity in vitro into adipocyte, osteoblast and chondroblast lineages [77]. After exposure to high doses of chemotherapeutic taxanes as paclitaxel, which act by stabilizing the b-subunit of tubulin in microtubules, MSCs seems to be able to uptake the drug and to deliver the antineoplastic agents at the tumor site, thus directly reducing tumor proliferation rates.

Many different methods of drug delivery have been described in the last decade, among which are immunoconjugates for targeting tumor-specific antigens, nanotools and genetically modified stem cells. Nevertheless, non-modified MSCs are probably the best option for anticancer drug local delivery, as they can rapidly adapt themselves to culture conditions and become home to pathological tissues when injected in vivo in addition to their own antineoplastic activity. MSCs, as well as extracellular vesicles (EVs) obtained from MSCs (MSCs-EV), can release active soluble factors and play an effective immunomodulatory role [78,79,80,81].

MSCs-EV may also represent a potential new Drug Delivery System (DDS) of therapeutic molecules. The main features which identify EVs as promising therapeutic candidates as DDS tools are essentially their high editability and low immunogenicity. The possibility of tumor growth blockades by exploiting cancer messengers is not only intriguing from a scientific perspective, but is also of clinical relevance since it allows a complete bioavailability of therapeutic compounds. Moreover, unlike other drug delivery platforms specifically targeting the cell surface, e.g., (monoclonal antibodies and peptide-based nanocarriers), EVs assure a much more precise and complex vehiculation system. However, it should be remarked that a careful evaluation of the TME, the choice of an ad hoc delivery tool as well as the electric charge of the nanocarrier have to be taken into account to obtain the most efficient results [82,83,84].

### 3.3. Implanted Biocompatible Scaffolds and Biomaterials

Tissue bioengineering has profoundly revolutionized recent therapeutic offers, mainly for local approaches to cancer progression. It involves interdisciplinary research, including biomaterial design and processing, surface characterization, and functionalization for improved cell-material interactions and imaging. Implanted scaffolds and injected hydrogels are two typical biomaterials that provide mechanical structures to tissue constructions, whether cells are suspended within or adhered to a three-dimension hydrogel framework [85]. Cells and tissues obtained and manipulated in vitro are then implanted into the patient, thus restoring compromised biological functions without having to resort to transplantation of biologic material from stranger donors. The study and definition of the material used is one of main fields of research. Natural materials contain in their structure, in specific signal sequences, all the information that can promote cell adhesion and maintain cell functions. The main strategies of tissue engineering involve: (i) use of molecules and growth factors that can induce tissue formations. They are mainly represented by: adhesion cell molecules, (integrins, cadherins, immunoglobulins), adhesion proteins to the extracellular matrix (selectins, fibronectin, laminin, tenascin C) and growth factors such as insulin-like growth factors (IGFs), fibroblasts growth factors (FGFs), transforming growth factors-β (TGFs-β), epidermal growth factors (EGFs), nerval growth factors (NGFs) and erythropoietin; (ii) use of isolated (stem) cells; (iii) use of cells sown into matrices or incorporated into them [86]. With respect to cancer, tissue-engineering can be efficiently exploited to generate reliable tumor models for vehicle drugs and target resistance to therapies [87]. Biomaterial scaffolds can efficiently deliver cancer immunotherapeutic agents to tumor masses such as vaccines, immunomodulators, and immune cells. Moreover, these approaches can be efficiently exploited for combinatorial therapies including conventional systemic chemotherapy [88]. Although no data are till now available regarding blockades of immune/inflammatory cascades, intrapleural polymeric films containing cisplatin have already been tested with promising results in MPM animal models [89], and a synergy between the two molecules has been demonstrated [90].

## 4. Conclusions

In addition to being a rare disease, MPM is classified as an orphan disease, of little interest to the pharmaceutical industry that does not invest sufficiently in this pathology.

This implies that the current therapeutic approaches against this tumor are non-specific and therefore poorly effective, resulting in a very limited life expectancy for the patient. There is, thus, an urgent need to find novel therapeutic strategies. Although few data are, till now, available on MPM, the specific disease features make it potentially susceptible to local therapeutic approaches, taking advantage of progressions in cellular manipulation and bioengineering. Innovative methodologies are landing the clinical scenario and the main results will derive from the pharmacological interaction with the active micro-environment, which surrounds neoplastic masses in the pleural space. The identification of patients likely to respond to specific therapeutic approaches (local vs. systemic and/or combinatorial strategies) needs clinical validation studies and requires a constant and productive interaction among the professionals with a complementary multidisciplinary background. Results from future trials could help the fight against MPM (and overall pleural malignancies) by proposing innovative local therapeutic approaches based on the design of drugs able to treat these still incurable tumors.

## Figures and Tables

**Figure 1 ijms-22-09014-f001:**
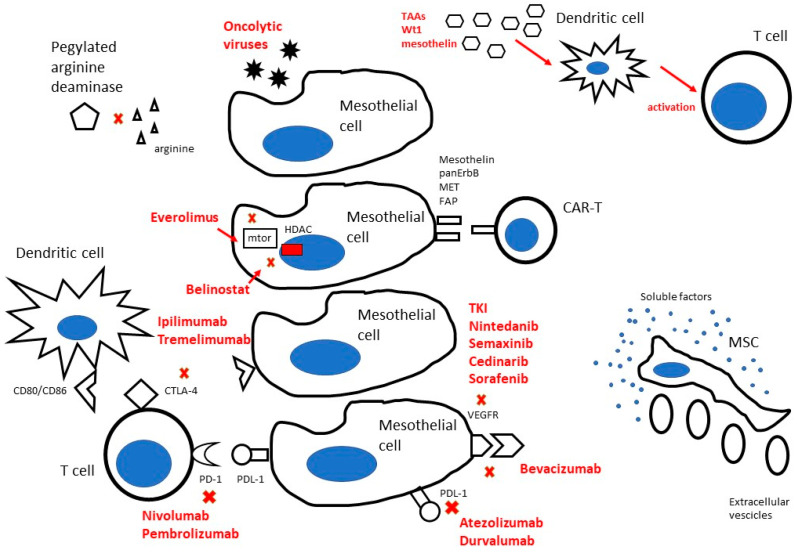
Main local therapeutic strategies for the management of malignant pleural mesothelioma. CAR-T, chimeric antigen receptor T-cell; CTLA-4, Cytotoxic T-Lymphocyte Antigen 4; FAP, Fibroblast Activation Protein; HDAC, histone deacetylase; MSC, mesenchymal stromal cell; mTor, mammalian target of rapamycin; PD-1, Programmed cell death protein 1; PDL-1, Programmed cell death LIGAND protein 1; VEGFR, vascular endothelial growth factor.

**Figure 2 ijms-22-09014-f002:**
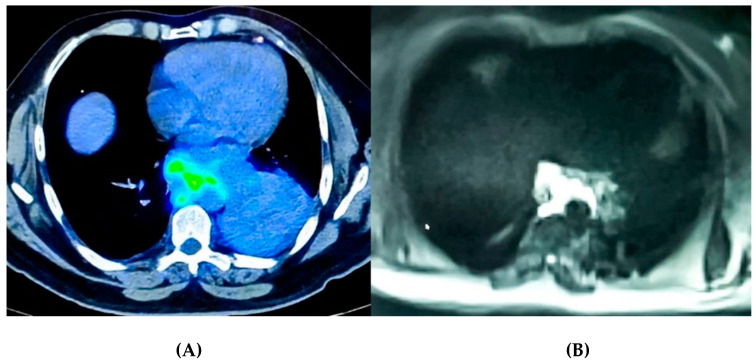
Matching functional and morphological imaging data. (**A**) CT/PET image showing a marked uptake of the tracer by a lymph node mass (with poor tissue uptake in the costo-vertebral shower). (**B**) IVIM sequence performed on the same patient. Comparing the two images, it can be observed that even if the CT/PET scan allows a greater morphological detail, the IVIM sequence has an equal capacity to highlight, with hyper-intensity of the signal, the same functional aspects.

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
