# Peer review of "Local Therapies and Modulation of Tumor Surrounding Stroma in Malignant Pleural Mesothelioma: A Translational Approach"

_ijms, 2021, doi:10.3390/ijms22169014_

Round 1

Reviewer 1 Report

The abstract is supposed to highly summarize the most important information. However, it failed to does its job by providing lots of very vague info. A few examples are given:

  1. Of course the 2nd line therapy is going to be used if the 1st line therapy fails. This is not a mesothelioma-specific thing. This applies to almost every disease treatment in medicine. 
  2. What exactly is unique about the mesothelioma TME? Besides claiming it is unique, I don’t have any further info. At least summarize what’s so special about the TME if it is needed to be summarized in abstract. 
  3. All the strategies listed from line 89-95 are also very vague. It felt like the authors are trying to convey some info about mesothelioma treatment, but failed to deliver any specific information. 

Overall in the main text, there are several key issues existing throughout the manuscript:

  1. There are many very vague claims, and the contents are not organized in a very logical fashion. The purpose of a review paper is to summarize in a logical fashion to convey useful information with a central theme. 
  2. There are many statements lasting for an entire paragraph without a single citation. 
  3. There are many false claims.
  4. There are many places where the contents are repeated without adding new info to the readers. For example in line 47-49, “an orphan disease in EU”, “a rare disease”, “an orphan disease” are practically the same thing but were repeated 3 times in 2 sentences. 
  5. Every tumor is made with malignant cells plus stroma and there are always dynamic cross-talks in the two components. Section 2.1 is currently a random list of information about the TME that seems to be very common in many types of cancers. I would highlight what’s unique about mesothelioma, rather than piling up random info.  
  6. Section 2.2: This is not really about imaging as a tool to assess TME. It is a far-stretch between TME and diagnostic imaging tools. If nothing has been reported about estimating CD8+ infiltration in mesothelioma patients, and the associations between IVIM and angiogenesis, then this section is pointless. As far as I know, estimation of CD8+ infiltration is still under development by City of Hope Cancer Centre and a few other sites, but nothing has really been applied to clinical practice. Figure 2 is pointless to me. PET CT shows great details. If IVIM is not superior, why bother? Several previous studies have compared and contrasted CT, MRI and PET. I would be more precise about what is stated if imaging is still part of the review. 
  7. Section 3: Why is nanoparticle its own section? Liposomes are not new. What is special about mesothelioma tx using liposomes? 
  8. Section 4: Again, there are so many statements without accompanying citations. The entire section is just a laundry list of random info about CAR-T with vague info about some trials. Plus, it is a very strange combination to put DC vaccine and CAR-T into the same sub-section. The immunology of DC vaccine was not accurately stated. MSC-related therapy sub-section is also a laundry list. Plus, MSC carries potential effects into supporting immunosuppression, and has been in fact tested for the treatment of transplant rejection by University of Michigan and Emory University. From my understanding, MSC-derived vesicles are “immunosuppressive” rather than “immunomodulatory”. The papers cited for this section have very questionable quality. Section 5 has similar issues to section 4. 
  9. I would suggest proof-reading the manuscript by a native speaker, as there are many noticeable grammar mistakes and odd phrases throughout the manuscript. Some terms are not used in the most well-accepted ways. For example, malignant cells are pretty much the same as transformed cells, but usually you don’t say malignant transformed cells, which is redundant.

Author Response

The abstract is supposed to highly summarize the most important information. However, it failed to does its job by providing lots of very vague info. A few examples are given:

We thank the Reviewer for careful revision of our work, which is now better in terms of quality and scientific message. Below the point-by-point answers (A).

  1. Of course the 2nd line therapy is going to be used if the 1st line therapy fails. This is not a mesothelioma-specific thing. This applies to almost every disease treatment in medicine. 

A1) We thank the Reviewer for point out this issue. Indeed the “mesothelioma-specific thing” is the concept that is summarized in the following sentence: “However, there are no approved agents”… as second line therapies. This fact reflects the definition of MPM as orphan disease.

  1. What exactly is unique about the mesothelioma TME? Besides claiming it is unique, I don’t have any further info. At least summarize what’s so special about the TME if it is needed to be summarized in abstract. 

A2) We thank the Reviewer for this comment. Although many published papers are already available regarding the MPM tumor microenvironment (e.g. Hiltbrunner S, Mannarino L, Kirschner MB, Opitz I, Rigutto A, Laure A, Lia M, Nozza P, Maconi A, Marchini S, D'Incalci M, Curioni-Fontecedro A, Grosso F. Tumor Immune Microenvironment and Genetic Alterations in Mesothelioma. Front Oncol. 2021 Jun 23;11:660039. doi: 10.3389/fonc.2021.660039. PMID: 34249695; PMCID: PMC8261295; Cheng L, Li N, Xu XL, Mao WM. Progress in the Understanding of the Immune Microenvironment and Immunotherapy in Malignant Pleural Mesothelioma. Curr Drug Targets. 2020;21(15):1606-1612. doi: 10.2174/1389450121666200719011234. PMID: 32682370; Napoli F, Listì A, Zambelli V, Witel G, Bironzo P, Papotti M, Volante M, Scagliotti G, Righi L. Pathological Characterization of Tumor Immune Microenvironment (TIME) in Malignant Pleural Mesothelioma. Cancers (Basel). 2021 May 24;13(11):2564. doi: 10.3390/cancers13112564. PMID: 34073720; PMCID: PMC8197227; Désage AL, Karpathiou G, Peoc'h M, Froudarakis ME. The Immune Microenvironment of Malignant Pleural Mesothelioma: A Literature Review. Cancers (Basel). 2021 Jun 26;13(13):3205. doi: 10.3390/cancers13133205. PMID: 34206956; PMCID: PMC8269097; Fusco N, Vaira V, Righi I, Sajjadi E, Venetis K, Lopez G, Cattaneo M, Castellani M, Rosso L, Nosotti M, Clerici M, Ferrero S. Characterization of the immune microenvironment in malignant pleural mesothelioma reveals prognostic subgroups of patients. Lung Cancer. 2020 Dec; 150:53-61. doi: 10.1016/j.lungcan.2020.09.026. Epub 2020 Oct 2. PMID: 33065463 ;…) the section 2.1 is entirely focused on the analysis on MPM surrounding stroma as actionable target by local approach. 

  1. All the strategies listed from line 89-95 are also very vague. It felt like the authors are trying to convey some info about mesothelioma treatment but failed to deliver any specific information. 

A3) We thank the Reviewer for this comment. However we would like to underline that the introduction sentences from line 89 to 95 summarize the rationale for local therapeutic approaches which are described in detail the whole paper. Coherently to Reviewer suggestions the sentences have been re-written as follows: “The present work aims at summarizing the novel approaches for local therapy in MPM, with primary focus on the rationale of this approach, which assures a closed contact with the immune/inflammatory and vascular processes which specifically characterized the MPM stroma”.

Overall in the main text, there are several key issues existing throughout the manuscript:

  1. There are many very vague claims, and the contents are not organized in a very logical fashion. The purpose of a review paper is to summarize in a logical fashion to convey useful information with a central theme. 
  2. There are many statements lasting for an entire paragraph without a single citation. 
  3. There are many false claims.

A 1-3) We thank the reviewer for critical reading of the manuscript. However, we disagree with most of the Reviewer comments. We have reorganized the review structure and added more references to exclude definitively that we wrote false claims. The review is logically structured as follows: 1) description of molecular features of MPM surrounding stroma to explain why it is appealing for local targeted therapeutic approach; 2) how the imaging tools can be exploited to characterize the MPM stroma in a more translational perspective; 3) how to local target MPM: where we are going; 4) conclusions.

  1. There are many places where the contents are repeated without adding new info to the readers. For example in line 47-49, “an orphan disease in EU”, “a rare disease”, “an orphan disease” are practically the same thing but were repeated 3 times in 2 sentences. 

A4) We thank the Reviewer for care reading of the text. However the term “rare” and “orphan” are not synonymous. Rare diseases are diseases which affect a small number of people compared to the general population (Orphanet: About rare diseases); Orphan disease is: a disease that has not been adopted by the pharmaceutical industry because it provides little financial incentive for the private sector to make and market new medications to treat or prevent it. An orphan disease may be a rare disease (according to US criteria, a disease that affects fewer than 200,000 people) or a common disease that has been ignored (such as tuberculosischolera, typhoid, and malaria) because it is far more prevalent in developing countries than in the developed world (https://www.medicinenet.com/). Moreover the definition of MPM as orphan disease by EU assures that the disease is enclosed into the European Reference Network (ERN) for the lung that is  dedicated to rare respiratory diseases which main aims are: to facilitate improvements in access to diagnosis and delivery of highquality, accessible and cost-effective healthcare for people living with rare diseases – act as focal points for medical training and research, information dissemination and evaluation, especially for rare diseases. This point has been added in the text.

  1. Every tumor is made with malignant cells plus stroma and there are always dynamic cross-talks in the two components. Section 2.1 is currently a random list of information about the TME that seems to be very common in many types of cancers. I would highlight what’s unique about mesothelioma, rather than piling up random info.  

A5) We thank the Reviewer for care reading of the text. However the section is not made by piling up random info. Every sentence written in it refers -specifically- to MPM, as indicated by the quotations. As already underlined by the Reviewer, every tumor (and - thus -  MPM) is made with malignant cells plus stroma and it is somehow expected that some biological processes could be active in other malignant settings, but in the text the clearly emerges the unique features of MPM surrounding stroma. The main and specific issue is related to the role played by asbestos (rather than activation of specific oncogenes) in both driving malignant transformation and modulating tumor surrounding stroma (for review see: Obacz J, Yung H, Shamseddin M, Linnane E, Liu X, Azad AA, Rassl DM, Fairen-Jimenez D, Rintoul RC, Nikolić MZ, Marciniak SJ. Biological basis for novel mesothelioma therapies. Br J Cancer. 2021 Jul 5. doi: 10.1038/s41416-021-01462-2. Epub ahead of print. PMID: 34226685; Urso L, Cavallari I, Sharova E, Ciccarese F, Pasello G, Ciminale V. Metabolic rewiring and redox alterations in malignant pleural mesothelioma. Br J Cancer. 2020 Jan;122(1):52-61. doi: 10.1038/s41416-019-0661-9. Epub 2019 Dec 10. PMID: 31819191; PMCID: PMC6964675; Abbott DM, Bortolotto C, Benvenuti S, Lancia A, Filippi AR, Stella GM. Malignant Pleural Mesothelioma: Genetic and Microenviromental Heterogeneity as an Unexpected Reading Frame and Therapeutic Challenge. Cancers (Basel). 2020 May 7;12(5):1186. doi: 10.3390/cancers12051186. PMID: 32392897; PMCID: PMC7281319, Stella GM. Carbon nanotubes and pleural damage: perspectives of nanosafety in the light of asbestos experience. Biointerphases. 2011 Jun;6(2):P1-17. doi: 10.1116/1.3582324. PMID: 21721837. …)

  1. Section 2.2: This is not really about imaging as a tool to assess TME. It is a far-stretch between TME and diagnostic imaging tools. If nothing has been reported about estimating CD8+ infiltration in mesothelioma patients, and the associations between IVIM and angiogenesis, then this section is pointless. As far as I know, estimation of CD8+ infiltration is still under development by City of Hope Cancer Centre and a few other sites, but nothing has really been applied to clinical practice. Figure 2 is pointless to me. PET CT shows great details. If IVIM is not superior, why bother? Several previous studies have compared and contrasted CT, MRI and PET. I would be more precise about what is stated if imaging is still part of the review. 

A6) we thank the Reviewer for these suggestions. We would like underline the following points:

If nothing has been reported about estimating CD8+ infiltration in mesothelioma patients… then this section is pointless. Even if no reports about estimating CD8+ infiltration in mesothelioma patients are available several articles reported the utility in other types of cancer; we bring to your attention reference number 3 (Sun et al. Lancet 2018).

In our review we report treatment not yet applied into clinical practice (e.g MesoCancerVa and DENIM trials) and for the same reason we think it’s correct to report an imaging promising field of research still not into clinical practice.

  • If nothing has been reported about … the associations between IVIM and angiogenesis, then this section is pointless. We are not aware of the fascinating research from the City of Hope Cancer Centre but several groups reported associations between IVIM and angiogenesis both in animal models and in several tumoral hystotype in humans. It has, so far, not been reported for MPM but we think this section is not pointless since it focuses a very advanced, active and promising field of research.

We add two reference demonstrating the associations between IVIM and angiogenesis in tumor other than MPM.

  • Figure 2 is pointless to me. PET CT shows great details. Looking at the image from a purely morphologic point of view may underestimate the amount of data and the potentiality of IVIM. IVIM is a quantitative method able to extract more than 15 quantitative indexes while PET-CT has several problems in standardizing the single quantitative index available (SUV).
  • If IVIM is not superior, why bother? Several previous studies have compared and contrasted CT, MRI and PET. Diagnostic “superiority” may not, in our opinion, be the only factor to be taken not account. PET-CT do use ionizing radiation and use radioactive tracers and it’s even more expensive and less available on the territory than MR. MR with IVIM do NOT use gadolinium contrast agents and do NOT use ionizing radiations; it’s more available than PET-CT and in several study has demonstrated “non inferiority” with PET-CT. Of course, the use of PET-CT is far more rooted into clinical practice; for how long it’s a matter of debate. I bring to your attention an commentary from the lancet 2019 on MR for lung tumor staging in comparison with PET-CT [Meyer M, Budjan J. Whole-body MRI for lung cancer staging: a step in the right direction. Lancet Respir Med. 2019 Jun;7(6):471-472].

  1. Section 3: Why is nanoparticle its own section? Liposomes are not new. What is special about mesothelioma tx using liposomes? 

A7) We thank the Reviewer for this critical comment. We have added in the conclusion of the section that the novelty is relative. However we are quite surprised by it since although liposomes are not really “new”, this approach has not fully reached the clinical scenario and many readers can not be aware of the therapeutic opportunity which they represent. Moreover, the “special” thing is that the local progressive pattern, specific of MPM, is an ideal model to use nanoparticles as vehicle to local deliver of drugs (standard chemo or bio-molecular agents). Indeed, as specified in the text, MPM represents a model to understand how to local treat other malignant diseases which present with local growth (e.g. malignant pleural effusions secondary to other primary masses; peritoneal cancers,…)

  1. Section 4: Again, there are so many statements without accompanying citations. The entire section is just a laundry list of random info about CAR-T with vague info about some trials. Plus, it is a very strange combination to put DC vaccine and CAR-T into the same sub-section. The immunology of DC vaccine was not accurately stated. MSC-related therapy sub-section is also a laundry list. Plus, MSC carries potential effects into supporting immunosuppression, and has been in fact tested for the treatment of transplant rejection by University of Michigan and Emory University. From my understanding, MSC-derived vesicles are “immunosuppressive” rather than “immunomodulatory”. The papers cited for this section have very questionable quality. Section 5 has similar issues to section 4. 

A8) We thank the Reviewer for the comments. References were implemented, as requested (Li Lv, Jiangchao Huang, Haipeng Xi, Xiangyang Zhou: Efficacy and safety of dendritic cell vaccines for patients with glioblastoma: A meta-analysis of randomized controlled trials. Int Immunopharmacol 2020 Jun;83:106336. doi: 10.1016/j.intimp.2020.106336. Epub 2020 Mar 23; Kristian M. Hargadon: Strategies to improve the efficacy of Dendritic Cell-Based immunotherapy for Melanoma (2017). Frontier Immun. (8); 1594; Peter Brossart: Dendritic cells in vaccination therapies of malignant diseases. Transfusion and Apheresis Science 27 (2002) 183–186; van Gulijk M, Dammeijer F, Aerts J, Vroman H. Combination strategies to optimize efficacy of dendritic cell-based immunotherapy. Front Immunol. (2018) 9:2759. doi: 10.3389/fimmu.2018.02759; Garg AD, Coulie PG, Van den Eynde BJ, Agostinis P. Integrating next generation dendritic cell vaccines into the current cancer immunotherapy landscape. Trends Immunol. (2017) 38:577–93. doi: 10.1016/j.it.2017.05.006; Alexander Markov, Lakshmi Thangavelu, Surendar Aravindhan, Angelina Olegovna Zekiy, Mostafa Jarahian: Mesenchymal stem/stromal cells as avaluable source for the treatment of immune-mediated disorders. Stem Cell Research & Therapy (2021) 12:192; Qingyuan Zheng, Shuijun Zhang, Wen-Zhi Guo and Xiao-Kang Li: The Unique Immunomodulatory Properties of MSC-Derived Exosomes in Organ Transplantation. Frontier Immunol. 2021, (12); 659621)

The purpose of this review was to describe all the current local therapies for MPM, not focusing on the advanced cell therapy alone, for this reason probably the section can appear too vague. DC vaccine and CAR-T therapy have been put in the same subsection because both treatments fall within the adoptive cell therapy, unlike the second approach of adevanced cell therapy, described in the subsection 4.2. The immunology of the DC vaccine has been implemented and clarified in the manuscript. About the immunosuppression/immunomodulatory potential of MSCs and their extracellular vescicles, is now well known that this is a direct result of a harmonic synergy of MSC-released signaling molecules, as mediators, cytokines, and chemokines, that can effectively modulate the inflammatory responses and control the infiltration process that finally leads to a regulated tissue repair/healing or regeneration process, as described by the authors cited in the manuscript. Nevertheless this subsection aimed to underline the importance of the possibility to use MSCs and MSCs-EV as carrier of terapeutic molecules, for example chemoterapeutic drugs. To date the studies on mSCs and MSCs-EV as drug delivery systems are limited, but this strategy could provide some advantages, mainly in an anti-tumor context, as the capacity to deliver drug directly near the tumor mass, making therapy more effective and requiring a smaller amount of drug.

  1. I would suggest proof-reading the manuscript by a native speaker, as there are many noticeable grammar mistakes and odd phrases throughout the manuscript. Some terms are not used in the most well-accepted ways. For example, malignant cells are pretty much the same as transformed cells, but usually you don’t say malignant transformed cells, which is redundant.

A9) The paper has been revised as required and redundant sentences has been removed.

Reviewer 2 Report

The Review article by Lisini et al., entitled “Local therapies and modulation of tumor surrounding stroma in malignant pleural mesothelioma: a translational approach”, summarizes the biologic effects of the current local therapies for MPM, including cell therapies and the mechanisms of interaction with the tumor micro-environment. In this Review, the authors deepen all the available local therapies applied or applicable to the MPM. They suggest a combinatorial strategy, using different local therapeutic approaches, as novel effective therapy against MPM.

This Review, provides a useful and detailed guide on the several strategies for local drug delivery in MPM. Although it is well expanded and well written, there are small paragraphs/parts to review:

  • The authors should reduce the caption for figure 1, adding the detailed description of local therapeutic strategies in the main text (2.1 chapter). A caption to the figure as long as the paragraph itself is not appropriate.
  • The authors should specify the acronym IVIM the first time it appears in the text. Please, do this at line 214 instead at line 215.
  • Please, correct the double subject at line 361: “MSCs they can release active soluble factors”.
  • Lastly, I would like to suggest to the authors to rewrite the paragraph of Conclusions. They should summarize first what the topic of the review was and then underlie the importance of finding new therapies for this rare and orphan disease. The ambition and the desire of the authors to research for new effective therapies, for example by combining and further developing the already existing therapies (synergistic effect), should be put as a secondary effect/future perspective in this type of article. Therefore, authors should rephrase the sentence: " By conducting an integrated translational and clinical research effort, our MPM team network has the ambition to find novel effective therapeutic strategies against MPM, an attempt which, to date, has been hampered by a lack of targeted therapies in front of chemo-induced systemic toxicities " by moving and integrating it to the end of the paragraph.

Author Response

The Review article by Lisini et al., entitled “Local therapies and modulation of tumor surrounding stroma in malignant pleural mesothelioma: a translational approach”, summarizes the biologic effects of the current local therapies for MPM, including cell therapies and the mechanisms of interaction with the tumor micro-environment. In this Review, the authors deepen all the available local therapies applied or applicable to the MPM. They suggest a combinatorial strategy, using different local therapeutic approaches, as novel effective therapy against MPM.

This Review, provides a useful and detailed guide on the several strategies for local drug delivery in MPM. Although it is well expanded and well written, there are small paragraphs/parts to review:

We really thank the reviewer for the careful revision of our manuscript which is now improved in its scientific value and impact. Below point-by-point answers (A).

  • The authors should reduce the caption for figure 1, adding the detailed description of local therapeutic strategies in the main text (2.1 chapter). A caption to the figure as long as the paragraph itself is not appropriate.
  1. We agree with this suggestion. The great part of the caption has been removed and added in the 2.1 section.
  • The authors should specify the acronym IVIM the first time it appears in the text. Please, do this at line 214 instead at line 215.
  1. We thank the Reviewer for careful revision and modified the text accordingly
  • Please, correct the double subject at line 361: “MSCs they can release active soluble factors”.
  1. We thank the Reviewer for careful revision and modified the text accordingly

  • Lastly, I would like to suggest to the authors to rewrite the paragraph of Conclusions. They should summarize first what the topic of the review was and then underlie the importance of finding new therapies for this rare and orphan disease. The ambition and the desire of the authors to research for new effective therapies, for example by combining and further developing the already existing therapies (synergistic effect), should be put as a secondary effect/future perspective in this type of article. Therefore, authors should rephrase the sentence: " By conducting an integrated translational and clinical research effort, our MPM team network has the ambition to find novel effective therapeutic strategies against MPM, an attempt which, to date, has been hampered by a lack of targeted therapies in front of chemo-induced systemic toxicities " by moving and integrating it to the end of the paragraph.
  1. We agree with Reviewer’s suggestion and modified the conclusion section as suggested.

Round 2

Reviewer 1 Report

I'm afraid the authors only made superficial changes. The review paper still lacks enough substance to be considered for publication. The references are not even in consistent format, which reflects that the authors did not really spend enough time and energy to improve.